# Tetherin Restricts SARS-CoV-2 despite the Presence of Multiple Viral Antagonists

**DOI:** 10.3390/v15122364

**Published:** 2023-11-30

**Authors:** Elena Hagelauer, Rishikesh Lotke, Dorota Kmiec, Dan Hu, Mirjam Hohner, Sophie Stopper, Rayhane Nchioua, Frank Kirchhoff, Daniel Sauter, Michael Schindler

**Affiliations:** 1Institute for Medical Virology and Epidemiology of Viral Diseases, University Hospital Tübingen, 72076 Tübingen, Germany; elena.hagelauer@med.uni-tuebingen.de (E.H.); rishikesh.lotke@med.uni-tuebingen.de (R.L.); dan.hu@med.uni-tuebingen.de (D.H.); mirjam.hohner@med.uni-tuebingen.de (M.H.); sophie.stopper@gmail.com (S.S.); daniel.sauter@med.uni-tuebingen.de (D.S.); 2Institute of Molecular Virology, Ulm University Medical Center, 89081 Ulm, Germany; dorota.kmiec@uni-ulm.de (D.K.); nchioua.rayhane@uni-ulm.de (R.N.); frank.kirchhoff@uni-ulm.de (F.K.)

**Keywords:** SARS-CoV, SARS-CoV-2, Tetherin, BST2, Spike, ORF7a, restriction factor

## Abstract

Coronavirus infection induces interferon-stimulated genes, one of which encodes Tetherin, a transmembrane protein inhibiting the release of various enveloped viruses from infected cells. Previous studies revealed that SARS-CoV encodes two Tetherin antagonists: the Spike protein (S), inducing lysosomal degradation of Tetherin, and ORF7a, altering its glycosylation. Similarly, SARS-CoV-2 has also been shown to use ORF7a and Spike to enhance virion release in the presence of Tetherin. Here, we directly compare the abilities and mechanisms of these two viral proteins to counteract Tetherin. Therefore, cell surface and total Tetherin levels upon ORF7a or S expression were investigated using flow cytometry and Western blot analysis. SARS-CoV and SARS-CoV-2 S only marginally reduced Tetherin cell surface levels in a cell type-dependent manner. In HEK293T cells, under conditions of high exogenous Tetherin expression, SARS-CoV-2 S and ORF7a reduced total cellular Tetherin levels much more efficiently than the respective counterparts derived from SARS-CoV. Nevertheless, ORF7a from both species was able to alter Tetherin glycosylation. The ability to decrease total protein levels of Tetherin was conserved among S proteins from different SARS-CoV-2 variants (α, γ, δ, ο). While SARS-CoV-2 S and ORF7a both colocalized with Tetherin, only ORF7a directly interacted with the restriction factor in a two-hybrid assay. Despite the presence of multiple Tetherin antagonists, SARS-CoV-2 replication in Caco-2 cells was further enhanced upon Tetherin knockout. Altogether, our data show that endogenous Tetherin restricts SARS-CoV-2 replication and that the antiviral activity of Tetherin is only partially counteracted by viral antagonists with differential and complementary modes of action.

## 1. Introduction

At the end of 2019, a new coronavirus emerged and spread rapidly around the world, leading the World Health Organization (WHO) to declare it a global viral pandemic on 11 March [1]. Its shared sequence identity (79.6%) with severe acute respiratory syndrome coronavirus (SARS-CoV) led to its designation as severe acute respiratory syndrome coronavirus 2 (SARS-CoV-2) [2]. SARS-CoV-2 belongs to the genus Betacoronavirus in the family of Coronaviridae and harbours a single-stranded, positive-sense RNA genome [3]. It encodes non-structural proteins (Nsps) and four structural proteins: nucleocapsid (N), membrane (M), envelope (E), and Spike (S) [4]. Accessory viral proteins such as ORF6 or ORF7a are frequently dispensable for viral replication in vitro but enable efficient replication in vivo by suppressing different branches of the host immune response [5].

SARS-CoV-2 infection induces an interferon (IFN) response, which subsequently triggers the expression of interferon-stimulated genes (ISGs) [6,7]. One protein that is induced in most cell types upon exposure to type I IFN is Tetherin (also known as bone marrow stromal antigen 2, BST2, or CD317) [8]. It has a unique topology with two membrane anchors: an N-terminal transmembrane domain and a C-terminal glycosylphosphatidylinositol (GPI) anchor [9]. These anchors are connected via a glycosylated coiled-coil ectodomain. As a dimer that is linked by disulfide bonds, Tetherin can restrict the release of a variety of enveloped viruses [10]. Antiviral activities of Tetherin have been demonstrated for retroviruses [11,12], rhabdoviruses [13], filoviruses [14], orthomyxoviruses [15], alphaviruses [16], herpesviruses [17], flaviviruses [18], and coronaviruses [19,20], among others. It is thought that this broad antiviral spectrum is achieved because Tetherin does not recognize specific viral motifs but acts as a physical intermembrane tether. In addition to its eponymous function, Tetherin has been shown to act as an innate immune sensor for viral infections that can induce a pro-inflammatory signalling response by directly activating the transcription factor NF-κB [21,22]. Furthermore, Tetherin-induced endocytosis of tethered viral particles leads to stimulation of endosomal pattern recognition receptors [23]. 

Many viruses encode Tetherin antagonists with different counteraction strategies. For example, HIV-1 Vpu degrades Tetherin [24,25], the glycoprotein of the Ebola virus has been shown to functionally inhibit Tetherin [26], while SIV Nef proteins and the HIV-2 envelope glycoprotein sequester Tetherin intracellularly in the trans-Golgi network [27,28,29]. For SARS-CoV, two viral proteins have been reported to counteract Tetherin: Spike redirects Tetherin to a lysosomal degradation pathway, leading to decreased Tetherin surface levels [30], while the accessory protein ORF7a binds Tetherin and inhibits its glycosylation, resulting in reduced functionality [31]. SARS-CoV-2 ORF7a was also shown to interact with Tetherin [20,32,33,34], and one report has described SARS-CoV-2 ORF7a as a Tetherin antagonist [20]. In addition, just recently, SARS-CoV-2 Spike and ORF3a have also been shown to contribute to the counteraction of Tetherin [35]. Here, we analyze the functional conservation of Tetherin antagonism by directly comparing the relative efficiency of S vs. ORF7a from SARS-CoV and SARS-CoV-2 and by exploring potential mechanistic differences. 

We employed functional tests to analyze the effects of the two proteins on surface and total Tetherin levels and determined SARS-CoV-2 replication kinetics in parental versus Tetherin-KO cells. Furthermore, we analyzed Spike protein orthologs from different SARS-CoV-2 variants to identify potential adaptive mutations. Altogether, our results support the functional role of Tetherin in the context of SARS-CoV-2 infection.

## 2. Materials and Methods

### 2.1. Cells and Cell Culture

Human embryonic kidney (HEK) 293T cells, HEK293T BOI66 MaMTH reporter cells, and cervical cancer-derived HeLa cells were cultivated in DMEM (Thermo Fisher Scientific, Waltham, MA, USA) supplemented with 10% fetal calf serum (FCS) and 100 μg/mL Penicillin/Streptomycin (P/S). Human colon cancer-derived epithelial Caco-2 cells were cultured in DMEM containing 10% FCS, 100 μg/mL P/S, and 1% non-essential amino acids. Human lung A549 cells were maintained in RPMI medium supplemented with 10% FCS and 100 μg/mL P/S. Caco-2 Tetherin knockdown cell lines were cultured in their regular cell culture medium that was supplemented with 2 μg/mL Blasticidin. All cells were grown in a humidified cell culture incubator at 37 °C and 5% CO_2_ and passaged two to three times a week. 

### 2.2. Generation of Caco-2 Tetherin Knockout Cell Lines

Caco-2 Tetherin knockout cells were generated by lentiviral transduction. Lentiviral stocks were produced in HEK293T cells. To this end, 0.7 × 10^6^ HEK293T cells (per well) were seeded in 6-well plates and transfected the next day with 0.45 μg pMD2G, 1.125 μg psPAX2, and 1.5 μg LentiCRISPRv2 blast CD317 (sgRNA sequence: GCTTCAGGACGCGTCTGCAG for KO1, GCTTACCACAGTGTGGTTGC for KO2) using PEI (transfection protocol for HEK293T cells, see below). At 24 h post-transfection, viruses were harvested. Caco-2 cells were seeded in a 6-well plate at a density of 0.3 × 10^6^ cells per well. The supernatant was aspirated one day later, and the cells were treated with the lentiviruses produced. Two days later, the cells were selected with 10 μg/mL blasticidin, and Tetherin KO was validated using Tetherin surface staining and flow cytometry. In addition, control and Tetherin KO cells were treated with IFNα (5 pg/mL), harvested after 24 h, and fixed with 2% PFA. After permeabilization with 0.2% saponin, the cells were stained with a PE-coupled anti-Tetherin antibody and analyzed using flow cytometry.

### 2.3. Viruses

Experiments involving replication-competent SARS-CoV-2 were conducted in a BSL3 laboratory. Recombinant SARS-CoV-2 reporter viruses expressing a fluorescent protein instead of ORF6 (SARS-CoV-2 ΔORF6-YFP) [36] or ORF7a (icSARS-CoV-2-mNG) [37] were previously described. IcSARS-CoV-2-mNG was obtained from the World Reference Center for Emerging Viruses and Arboviruses (WRCEVA) at the UTMB (University of Texas Medical Branch). Virus stocks were generated by infection and propagation in Caco-2 cells, and viral titers were determined by infection rates (fluorescent cells) after serial dilution. SARS-CoV-2 ΔORF6-YFP was a kind gift from Prof. Armin Ensser (Institute for Clinical and Molecular Virology, Friedrich-Alexander University Erlangen-Nürnberg (FAU), 91054 Erlangen, Germany).

### 2.4. Plasmids and Antibodies

pEYFP-C1 CD317 was described previously [38]. A pmCherry-C1 CD317 construct was created similarly by amplifying and inserting CD317 into pmCherry_C1 via BamHI/XhoI. Plasmids encoding the Spike proteins of SARS-CoV and SARS-CoV-2 were cloned into the pCG_IRES-GFP backbone using XbaI/MluI. Human codon-optimized plasmids obtained from GenScript for SARS-CoV S and from Markus Hoffmann (Infection Biology Unit, German Primate Center—Leibniz Institute for Primate Research, Göttingen, Germany) for SARS-CoV-2 S served as templates. Spike expression plasmids of SARS-CoV-2 variants were generated by amplifying the Spike cDNA sequences from expression plasmids kindly provided by Markus Hoffmann (Infection Biology Unit, German Primate Center—Leibniz Institute for Primate Research, Göttingen, Germany) and inserting them into the pCG_IRES-GFP vector via XbaI/MluI. In all S plasmids, the Kozak sequence GCCACC was introduced upstream of the gene. To generate a GFP-Spike fusion protein, SARS-CoV-2 Spike was amplified with primers introducing 5′-BamHI and 3′-MluI sites, thereby allowing an in-frame insertion into the GFP ORF of pWPXLd. Expression plasmids encoding SARS-CoV ORF7a C-V5 and SARS-CoV-2 ORF7a C-V5 were generated by subcloning inserts into XbaI and MluI restriction sites of the pCG_IRES eGFP vector. SARS-CoV-2 ORF7a was also PCR amplified with primers introducing XhoI/EcoRI restriction sites and cloned into the pmScarlet_C1 backbone. pCG_HIV-1 NL4-3_Vpu-IRES-GFP was described before [39]. MaMTH plasmids encoding Gal4, Pex7 and human Tetherin and SARS-CoV-2 Spike were reported before [40,41]. SARS-CoV-2 ORF7a was cloned into the MaMTH Prey vector using Gateway cloning technology (Thermo Fisher Scientific, Waltham, MA, USA). The correctness of new constructs was confirmed by Sanger sequencing (Eurofins Genomics, Ebersberg, Germany).

All antibodies used in this study were obtained commercially. Unconjugated antibodies: monoclonal mouse anti-SARS-CoV/SARS-CoV-2 Spike (BIOZOL, Eching, Germany; GTX632604), polyclonal rabbit anti-BST2 (Proteintech, Planegg-Martinsried, Germany; 13560-1-AP), polyclonal rabbit anti-V5 (Abcam, Cambridge, UK; ab9116), and monoclonal rat anti-GAPDH (BioLegend, San Diego, CA, USA; 607902). Conjugated antibodies: PE-conjugated human anti-BST2 (Miltenyi Biotec, Bergisch Gladbach, Germany; 130-101-656), APC-conjugated mouse anti-human CD317 (BioLegend, San Diego, CA, USA; 348410), goat-anti-mouse 680RD (LI-COR Biosciences, Lincoln, NE, USA; 926-68070), goat-anti-rabbit 800CW (LI-COR Biosciences, Lincoln, NE, USA; 926-32211), goat-anti-rabbit Alexa594 (Thermo Fisher Scientific, Waltham, MA, USA; A-11012), and goat-anti-rat 800CW (LI-COR Biosciences, Lincoln, NE, USA; 926-32219).

### 2.5. Transfection of HEK293T Cells

HEK293T cells were seeded one day prior to transfection in 6-well plates using 7.5 × 10^5^ cells per well. For transfection of one well, 2 μg plasmid DNA was diluted in 100 μL OptiMEM (Thermo Fisher Scientific, Waltham, MA, USA). The DNA mix was incubated with 100 μL OptiMEM containing 6 μL polyethylenimine (PEI) for 15 min at room temperature (RT). The reaction mix was added dropwise to the cells and incubated at 37 °C. After 16 h, the medium was changed, and the cells were incubated for another 48 h at 37 °C.

### 2.6. Transfection of HeLa and A549 Cells

HeLa and A549 cells were seeded one day prior to transfection in 12-well plates using 2 × 10^5^ cells per well. For transfection of one well, 100 μL jetPRIME buffer (Polyplus, Illkirch-Graffenstaden, France) was mixed with 1 μg DNA and vortexed. After spinning down the DNA mix, 2 μL jetPRIME reagent was added, and the mix was vortexed and spun down again. After 10 min of incubation at RT, the reaction mix was added dropwise to the cells and incubated at 37 °C. After 4 h, the medium was changed, and the cells were incubated for another 48 h at 37 °C.

### 2.7. Transfection of Caco-2 Cells

Caco-2 cells were seeded one day prior to transfection in 12-well plates using 2 × 10^5^ cells per well. For transfection of one well, 1 μg plasmid DNA was diluted in 100 μL OptiMEM (Thermo Fisher Scientific, Waltham, MA, USA). The DNA mix was incubated with 100 μL OptiMEM containing 3 μL Lipofectamine2000. After 5 min, the reaction mix was added dropwise to the cells and incubated at 37 °C. After 4 h, the medium was changed, and the cells were incubated for another 48 h at 37 °C.

### 2.8. Western Blot

At 48 h after the medium was changed following transfection, cells were harvested, washed with PBS, and lysed on ice for 30 min using RIPA buffer (10 mM Tris-HCl pH 7.4, 140 mM NaCl, 1% TritonX-100, 0.1% Na-deoxycholate, 0.1% SDS, 1 mM EDTA, 0.5 mM EGTA, Protease Inhibitor Cocktail Tablets, EDTA-Free (Sigma-Aldrich, St. Louis, MO, USA)). Cell lysate was spun down for 10 min at 20,000 g and 4 °C, and the supernatant was mixed with 6× SDS loading buffer and denatured at 95 °C for 5 min. Precleared cell lysates were analyzed using sodium dodecyl sulfate polyacrylamide gel electrophoresis (SDS-PAGE) utilizing 10% polyacrylamide gels. Separated proteins were transferred onto a polyvinylidene fluoride (PVDF) membrane by wet transfer. The membrane was blocked in 5% milk in TBS-T (0.1% Tween-20 in TBS) for at least 1 h at RT and subsequently incubated with the primary antibody (in 5% milk in TBS-T) at 4 °C overnight or at RT for 2 h. After washing the membrane three times for 10 min with TBS-T, the secondary antibody (in TBS-T) was incubated for 1 h at RT. The membrane was again washed three times with TBS-T, and fluorescence was imaged using the Odyssey^®^ Fc Imaging System (LI-COR Biosciences, Lincoln, NE, USA).

### 2.9. Infection Experiments

Caco-2 Tetherin knockdown cells and empty-vector transduced control cells were used for infection with SARS-CoV-2. One day prior to infection, 2 × 10^4^ cells per well were seeded into a 96-well flat bottom plate. The next day, cells were infected at different multiplicities of infection (MOIs) using infection medium (DMEM, 5% FCS, 100 μg/mL P/S). After 4 h, medium was changed, and viral replication and spread were monitored using live-cell imaging at 37 °C for 72 h.

### 2.10. Fluorescence Microscopy

Coverslips were coated with poly-L-lysine (0.01 mg/mL in PBS) for 1 h at 37 °C. After washing with PBS, 1.5 × 10^5^ HEK293T cells per well were seeded onto the coverslips in a 12-well format. The next day, cells were transfected to express fluorophore-conjugated proteins. After 24 h at 37 °C, cells were fixed with 2% PFA in PBS at RT for 15 min. Subsequently, the fixed cells were washed with PBS. To visualize nuclei, cells were stained with DAPI (1:20,000 in PBS) for 10 min at RT. After washing with PBS, the coverslips were transferred onto microscopy slides. The slides were dried overnight at 4 °C, and cells were imaged the next day with a fluorescence microscope (ZEISS, Oberkochen, Germany; ZEISS Axio). 

### 2.11. Immunofluorescence Staining and Flow Cytometry

For flow cytometric analysis, detached cells were resuspended in FACS buffer (1% FCS in PBS) and transferred to a 96-well plate. For cell surface staining, cells were resuspended in APC-conjugated or PE-conjugated Tetherin antibody dilution (1:11 in PBS) and stained for 30 min at 4 °C in the dark. Cells were washed with FACS buffer and stored at 4 °C until flow cytometric measurement using the MACSquant VYB cytometer (Miltenyi Biotec, Bergisch Gladbach, Germany). Since bicistronic plasmids encoding GFP via an IRES were used, transfected and untransfected cells were discriminated via their GFP expression. The side and forward scatter were used to gate for living (SSC-A vs. FSC-A) and further single cells (FSC-H vs. FSC-A). PE-A was used to gate for PE+, i.e., Tetherin-expressing cells. N-fold downregulation was calculated by dividing the mean fluorescence intensity (MFI) of PE in the GFP negative population by the MFI of PE in the GFP positive population. Data were analyzed using the FlowLogic 8.3 software.

### 2.12. Mammalian-Membrane Two-Hybrid (MaMTH) Assay

HEK293T B0166 Gaussia luciferase reporter cells were co-transfected in 96-well plates with 25 ng SARS-CoV-2 protein Bait and 25 ng human Tetherin or control protein Prey MaMTH vectors in triplicates using PEI transfection reagent. Gal4 transcription factor served as a positive control, whereas SARS-CoV-2 Bait proteins with Pex7 Prey were used as negative controls. The following day, Bait protein expression was induced with 0.1 μg/mL doxycycline. Cell-free supernatants were harvested 2 days post-transfection and the released Gaussia luciferase was measured 1 s after injecting 20 mM coelenterazine substrate using an Orion microplate luminometer.

### 2.13. Statistical Analysis

For comparisons of Tetherin levels in ORF7a or S expressing cells to control samples, one-way ANOVA was applied using Šídák’s multiple comparisons test. MaMTH assay results were analyzed by two-way ANOVA using Šídák’s multiple comparisons test. Alpha was set equal to 5.0%. Statistical tests were performed using the Graphpad Prism 9.4.1 software.

## 3. Results

### 3.1. Coronavirus S and ORF7a Marginally Reduce Cell Surface Tetherin in a Cell Type-Dependent Manner

To investigate if SARS-CoV- and SARS-CoV-2-derived S and ORF7a proteins modulate Tetherin levels at the plasma membrane, we transfected A549, Caco-2, HeLa, and HEK293T cells to express S or ORF7a together with GFP from bicistronic expression plasmids (Figure 1a). This allowed us to readily quantify Tetherin cell surface levels in the presence of either S or ORF7a via flow cytometry. The ability of the viral proteins to downregulate Tetherin varied between the four cell lines investigated and was generally not very pronounced, even though some of the effects reached statistical significance (Figure 1a–e). In the lung adenocarcinoma cell line A549, representing human alveolar basal epithelial cells (Figure 1b), both S proteins downregulated Tetherin about two-fold, whereas in the colon cancer-derived epithelial cell line Caco-2 none of the viral proteins showed a significant effect (Figure 1c). In contrast, in HeLa cells, only SARS-CoV-2 ORF7a had a very slight, albeit significant effect in reducing Tetherin cell surface levels (Figure 1d). To test the effect of basal Tetherin levels on Spike- and ORF7a-mediated downmodulation, we moved to HEK293T cells, which express low endogenous levels of Tetherin [12] and enable an easy titration of Tetherin levels via transfection. Low levels of endogenous Tetherin at the cell surface of HEK293T cells were reproducibly downregulated by SARS-CoV and SARS-CoV-2 S (Figure 1e), similar to the phenotype observed in A549 cells (Figure 1b). Similarly, when HEK293T cells were transfected to express Tetherin, we witnessed a modest reduction in cell surface Tetherin by the coronavirus proteins, even though effects failed to reach significance due to high variations in this assay (Figure 2a,b). Furthermore, Tetherin downregulation by S and ORF7a was clearly less efficient than that exerted by HIV-1 Vpu, an established and highly potent Tetherin antagonist [11]. Together, S and ORF7a from SARS-CoV and SARS-CoV-2 showed weak and cell-line-dependent abilities to downregulate cell surface Tetherin. More specifically, statistically significant reductions in Tetherin surface levels were mostly attributable to S rather than ORF7a. This indicates that S and ORF7a act differentially on Tetherin and that the potential antagonizing activity of both proteins on Tetherin is not based on lowering cell surface levels of the restriction factor.

### 3.2. SARS-CoV-2 S and ORF7a Reduce Total Cellular Tetherin Levels

Thus far, we analyzed the ability of the viral proteins to reduce levels of cell surface Tetherin; however, SARS-CoV and SARS-CoV-2 bud into intracellular vesicles (5, 6) and may therefore be trapped and restricted by Tetherin intracellularly. Furthermore, there could be differential effects of viral antagonists on de novo synthesized versus cell surface Tetherin. Hence, the established HEK293T Tetherin expression model was used, but instead of measuring Tetherin at the cell surface, cellular lysates were prepared and total Tetherin levels were analyzed and quantified using Western blot (WB) (Figure 2c,d). Using this approach, we found that SARS-CoV-2 S and ORF7a, and to a lesser extent SARS-CoV S, significantly reduced total cell-associated protein levels of Tetherin (Figure 2c,d). Of note, the ORF7a proteins of both viruses also affected the migration pattern of Tetherin (Figure 2c). Since the ORF7a protein of SARS-CoV is known to alter Tetherin glycosylation and causes similar changes in the band profile of Tetherin in WB [31], this may suggest that ORF7a of SARS-CoV-2 interferes with the glycosylation of Tetherin in a similar manner. 

Similarly, we tested the ability of SARS-CoV-2 S variants (α, γ, δ, ο in comparison to the Wuhan Hu-1 reference strain) to antagonize Tetherin. This revealed that S proteins derived from all tested variants retained the ability to significantly reduce total Tetherin levels (Figure 2e,f), indicating that there is an ongoing selection pressure on this S protein activity.

### 3.3. SARS-CoV-2 S and ORF7a Colocalize with Tetherin but Only ORF7a Shows Evidence for Direct Interaction with the Restriction Factor

We next sought to assess a potential interaction between Tetherin and SARS-CoV-2 S or ORF7a. To this end, we transfected HEK293T cells to co-express Tetherin fused to mCherry or YFP, as well as either SARS-CoV-2 S fused to GFP or ORF7a fused to mScarlet. Then, cells were analyzed for protein localization using fluorescence microscopy. Upon co-expression of Tetherin and the two SARS-CoV-2 proteins, we detected pronounced areas of co-localization, indicating that S and ORF7a are in close proximity to Tetherin (Figure 3). 

To further elucidate if the viral proteins directly interact with Tetherin, we took advantage of the mammalian-membrane two-hybrid (MaMTH) assay (Figure 4). In this assay, the transcription factor Gal4 is released from a bait protein upon interaction with its prey due to the reconstitution of ubiquitin that is cleaved by deubiquitinating enzymes. Then, Gal4 activates luciferase expression in the nucleus (Figure 4a). Over-expression of Gal4 served as a positive control, while a random protein (Pex7) served as a negative control. As expected, expression of individual prey (Tetherin) or bait (S, ORF7a) constructs resulted in low luciferase reporter activity as compared to Gal4-only expression (Figure 4b). Similarly, luciferase signals were close to the Pex7 negative control when SARS-CoV-2 S bait was combined with Tetherin prey (Figure 4c). Importantly, however, the combination of ORF7a with Tetherin resulted in a robust luciferase signal that was comparable to that of the positive control Gal4 (Figure 4c). Hence, we conclude that while both SARS-CoV-2 S and ORF7a share areas of subcellular co-localization with Tetherin, only ORF7a shows evidence for a physical interaction with the antiviral factor in our two-hybrid assay.

### 3.4. Tetherin Restricts Replication of Authentic SARS-CoV-2

We next aimed to assess the importance of SARS-CoV-2 to maintain functional Tetherin antagonism in the context of viral infection of SARS-CoV-2 permissive cells. For this, we decided to monitor viral replication in the presence and absence of Tetherin. These experiments were conducted in Caco-2 epithelial cells since those cells sustain authentic SARS-CoV-2 replication without the necessity of overexpressing the entry receptors ACE2 or TMPRSS2. Furthermore, Caco-2 cells express intracellular endogenous Tetherin [43] that might hence localize to the internal compartments of SARS-CoV-2 assembly and release. Upon CRISPR/Cas9-mediated knock-out of Tetherin in these cells (Figure 5a–c), we infected two independent Tetherin KO Caco-2 cell lines with SARS-CoV-2 reporter viruses either expressing yellow fluorescent protein (YFP) instead of ORF6 or mNeonGreen (mNG) instead of ORF7. Here, we used bulk cultures of lentivirus-generated Tetherin KO Caco-2 cells to reduce potential off-target effects. We monitored viral replication and spread using live-cell imaging for 72 h (Figure 5d–g). Of note, in both Tetherin KO cell lines, we observed increased spread and replication of both reporter viruses compared to control cells, especially at ~24 h to ~48 h, at different MOIs (Figure 5d,e). This difference turned out to be significant for the SARS-CoV-2 ΔORF6-YFP infected cells when total fluorescence was integrated as area under the curve (AUC) as a proxy for overall viral replication (Figure 5f) and followed the trend for the SARS-CoV-2 ΔORF7-mNG reporter virus (Figure 5g). Altogether, endogenous Tetherin restricts replication of SARS-CoV-2 in epithelial cells, even in the presence of the Tetherin antagonists S and/or ORF7a.

## 4. Discussion

Coronaviruses including SARS-CoV-2 induce an interferon response that is counteracted at several levels [44,45]. One viral strategy is to encode antagonists of specific ISGs such as Tetherin, which otherwise restricts the release of a variety of enveloped viruses [11,12]. Examples of known viral Tetherin antagonists are the lentiviral accessory proteins Vpu and Nef [24,25,46,47,48], KSHV K5 [49,50], or the Ebola virus glycoprotein [26]. For SARS-CoV and SARS-CoV-2, accumulating evidence suggests that ORF7a and Spike are Tetherin antagonists [20,30,31,35]. Moreover, one recent report suggests that SARS-CoV-2 ORF3a counteracts Tetherin by sequestering it in late endocytic organelles [35]. Interestingly, a previous study suggests that the efficiencies and mechanisms of Tetherin counteraction vary between individual coronaviral antagonists. For example, SARS-CoV ORF7a has been shown to alter the glycosylation of Tetherin [31], while no such effect was observed for its SARS-CoV-2 ortholog [20,35]. Here, we therefore directly compared the anti-Tetherin activities of SARS-CoV and SARS-CoV-2 ORF7a and S and analyzed S proteins from different SARS-CoV-2 variants for their ability to counteract Tetherin. 

While SARS-CoV and SARS-CoV-2 S proteins slightly reduce cell surface levels of Tetherin in various cell lines, no such activity was observed for ORF7a, which is in line with differential and thus complementary modes of Tetherin antagonism by the two viral proteins (Figure 1). Furthermore, both S and ORF7a modulated total Tetherin levels, albeit to different extents. S and ORF7a of SARS-CoV only marginally, whereas SARS-CoV-2 S potently reduced total Tetherin protein levels, an activity that was also inherent to SARS-CoV-2 ORF7a (Figure 2a,b). In addition, ORF7a derived from SARS-CoV and SARS-CoV-2 altered the migration pattern of Tetherin in Western blot, indicating a possible interference with Tetherin glycosylation (Figure 2c). In conclusion, our data strengthen previous observations indicating that SARS-CoV and SARS-CoV-2 have evolved multiple Tetherin antagonists with differential modes of action. Furthermore, our data do not show major differences in the ability of SARS-CoV-2 variant-specific S proteins to reduce total cellular Tetherin levels (Figure 2e,f). Hence, Tetherin antagonism seems to be a conserved function of SARS-CoV-2 S. Whether ORF7a and ORF3a proteins derived from SARS-CoV-2 variants differ in their ability to counteract Tetherin is currently unknown and subject to further investigation. 

Another finding of our work is that endogenous Tetherin restricts SARS-CoV-2 replication. Of note, this phenotype was observed in the presence of the Tetherin antagonists S, ORF3a, and ORF7a, as shown by our live cell replication kinetics with SARS-CoV-2-ΔORF6-YFP (Figure 5d,f) in Caco-2 cells with and without Tetherin expression. This might highlight the need for SARS-CoV-2 to use additional mechanisms that blunt the interferon response, not only at the level of effector proteins but also during induction. Furthermore, SARS-CoV-2 lacking ORF7a has similar replication kinetics as compared to the variant expressing all three known Tetherin antagonists: ORF3a, ORF7a, and S. In conclusion, at least in our model, S and ORF3a—or yet another unknown mechanism of counteraction— can compensate for ORF7a to antagonize Tetherin, indicating a high pressure on viral evolution to maintain functional Tetherin antagonism.

Altogether, we here confirm and extend previous findings on Tetherin counteraction by SARS-CoV and the pandemic SARS-CoV-2. We demonstrate functional Tetherin antagonism by the S and ORF7a proteins of both viruses; moreover, there are differences in the mechanisms and efficiencies of Tetherin antagonism when comparing S and ORF7a of SARS-CoV and SARS-CoV-2. Whether the latter contribute to the higher transmissibility, spread, and pathogenicity of SARS-CoV-2 is an important question that warrants further investigation.

## Figures and Tables

**Figure 1 viruses-15-02364-f001:**
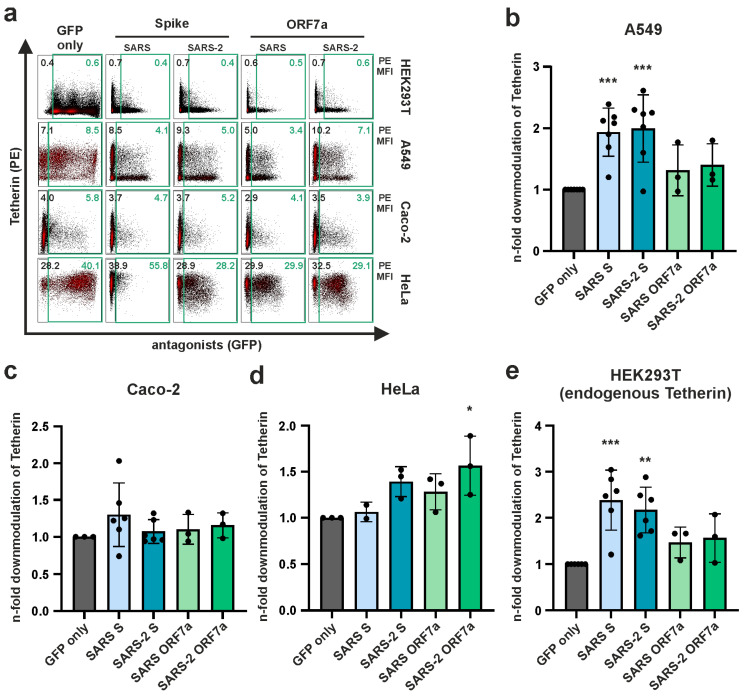
Effect of SARS-CoV and SARS-CoV-2 Spike (S) or ORF7a on endogenous Tetherin surface levels in different cell lines. (**a**) A549, Caco-2, HeLa, and HEK293T cells were transfected with expression plasmids encoding for SARS-CoV and SARS-CoV-2 S or ORF7a together with GFP from a bicistronic plasmid. Two days post-transfection, cells were harvested, surface-stained with a PE-conjugated anti-Tetherin antibody, and analyzed using flow cytometry. Representative plots of at least three independent biological replicates are shown. Numbers indicate mean PE fluorescence intensity of GFP-negative (black) and GFP-positive (green) populations; (**b**–**e**) Mean PE fluorescence of the GFP-negative population was divided by the mean PE fluorescence of the viral protein expressing GFP-positive population, and the n-fold change compared to the GFP-only control was calculated. Mean values of at least three independent biological replicates are shown. Error bars indicate SD. Statistical significance was tested by one-way ANOVA with Sidak’s multiple comparisons (* *p* ≤ 0.05, ** *p* ≤ 0.01, *** *p* ≤ 0.001).

**Figure 2 viruses-15-02364-f002:**
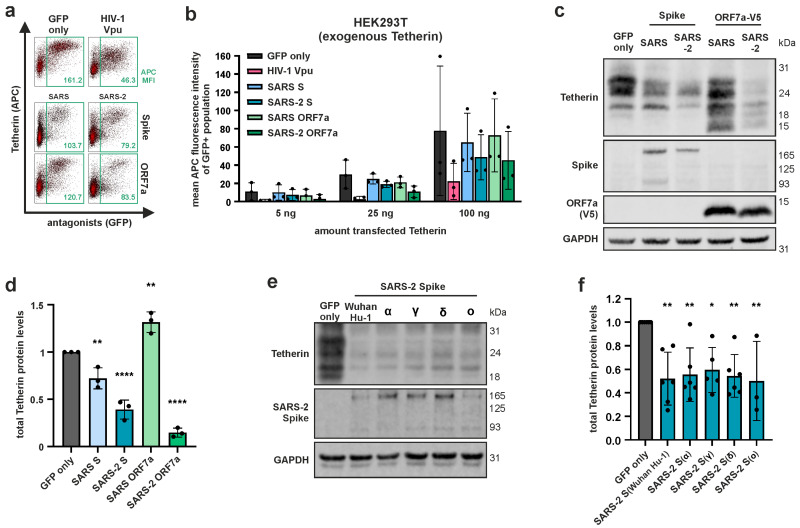
Effect of SARS-CoV or SARS-CoV-2 Spike (S) or ORF7a on total cellular Tetherin levels. (**a**,**b**) HEK293T cells were co-transfected with a plasmid encoding human Tetherin and bicistronic plasmids encoding for SARS-CoV and SARS-CoV-2 S or ORF7a protein together with GFP. (**a**) Representative plots for 100 ng Tetherin are shown. Numbers indicate mean APC fluorescence intensity of GFP-positive populations. (**b**) Increasing amounts of Tetherin expression plasmid (5, 25, 100 ng) were co-transfected. An expression plasmid for HIV-1 Vpu served as positive control. Two days post-transfection (d.p.t.), cells were harvested, surface-stained with an APC-conjugated anti-Tetherin antibody, and analyzed using flow cytometry. Mean APC fluorescence intensity of the GFP-positive population is shown. (**c**) Similar setup as in (**a**). Co-transfection of 62.5 ng Tetherin expression plasmid. Two d.p.t., cells were harvested and lysed. Total Tetherin protein levels in cell lysates were determined using Western blot. (**d**) Tetherin levels were quantified and normalized to GAPDH. Tetherin levels of the GFP-only control were set to 1. (**e**) HEK293T cells were co-transfected with expression plasmids encoding for Tetherin and S proteins of different SARS-CoV-2 variants. Two d.p.t., cells were harvested, and cell lysates were analyzed using Western blot. (**f**) Tetherin levels were quantified and normalized as described in (**b**). (**c**,**e**) Representative blots of three independent biological replicates each. (**d**,**f**) Mean values of at least three independent biological replicates. Error bars indicate SD. Statistical significance was tested by one-way ANOVA with Sidak’s multiple comparisons (* *p* ≤ 0.05, ** *p* ≤ 0.01, **** *p* ≤ 0.0001).

**Figure 3 viruses-15-02364-f003:**
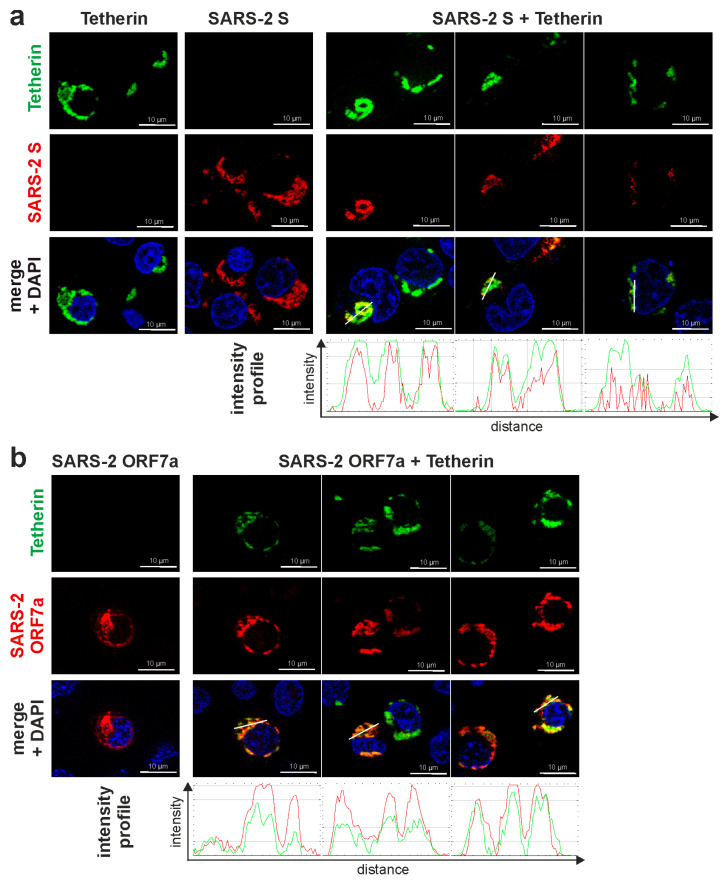
SARS-CoV-2 Spike (S) and ORF7a colocalize with Tetherin. HEK293T cells were transfected with expression plasmids encoding (**a**) SARS-CoV-2 S-GFP or (**b**) SARS-CoV-2 ORF7a-mScarlet and (**a**) Tetherin-mCherry or (**b**) Tetherin-YFP. For simplicity, Tetherin is shown in green and the viral proteins in red. 24 h post-transfection, cells were fixed with 2% PFA and stained with DAPI. Images were acquired using a fluorescent microscope and processed using ImageJ. For colocalization analysis, intensity profiles for the indicated lines are shown. Representative images of two independent experiments are shown (scale bar = 10 µm).

**Figure 4 viruses-15-02364-f004:**
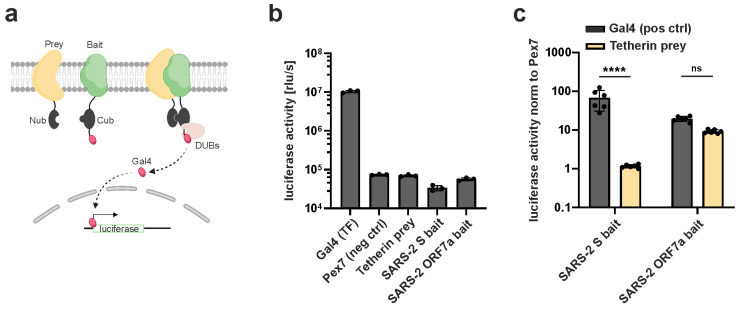
Mammalian-membrane two-hybrid assay (MaMTH) reveals interaction of SARS-CoV-2 ORF7a with Tetherin. (**a**) Schematic representation of the principle of the MaMTH assay. Bait and prey proteins are tagged with two inactive fragments of ubiquitin. Interaction of the bait and prey protein leads to reconstitution of an active ubiquitin that is cleaved by endogenous deubiquitinating enzymes (DUBs). This results in the release of the transcription factor Gal4 that activates reporter gene (Gaussia luciferase) transcription. The scheme was created using BioRender, adapted from Saraon et al. [42]. (**b**,**c**) HEK293T BOI66 reporter cells are transfected with prey and bait constructs. Expression plasmids for Gal4 and Pex7 served as positive and negative controls, respectively. After 48 h, cell culture supernatants were harvested and luminescence was measured. (**b**) Luciferase activities of controls are shown. (**c**) Tetherin was used as prey combined with SARS-2 S or ORF7a as bait. The transcription factor GAL4 was again added as a positive control. Luciferase activity was normalized to the Pex7 negative control. Mean values of at least three independent biological replicates are shown. Error bars indicate SD. Statistical significance was tested by one-way ANOVA with Sidak’s multiple comparisons (**** *p* ≤ 0.0001, ns: not significant).

**Figure 5 viruses-15-02364-f005:**
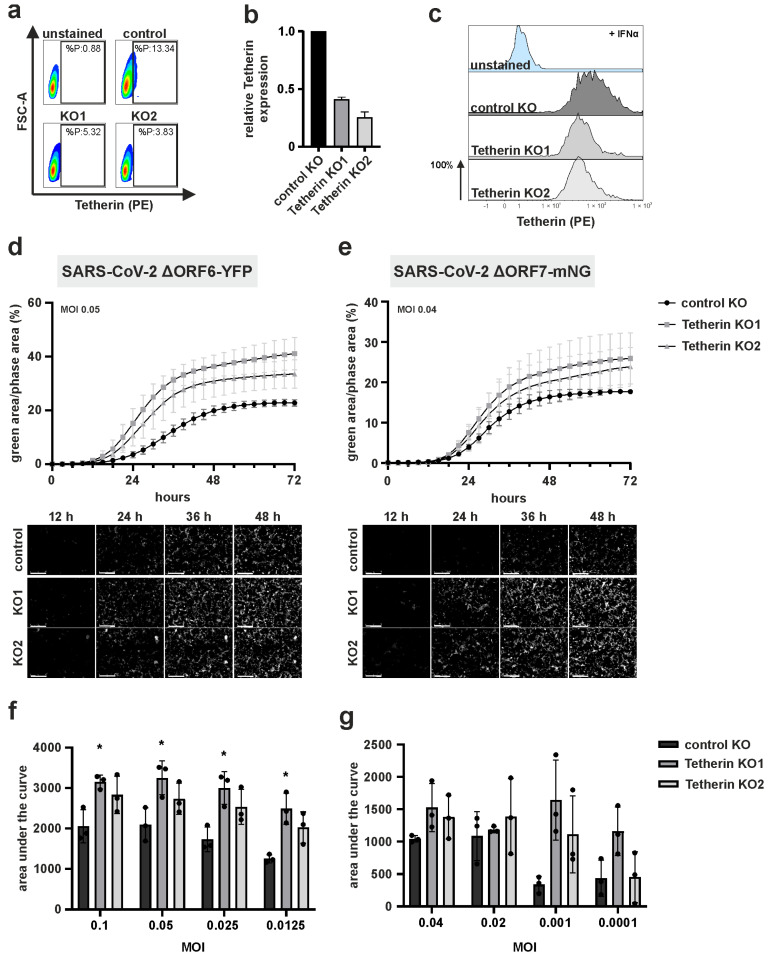
ORF6- and ORF7-deleted SARS-CoV-2 replication in Caco-2 cells is enhanced upon Tetherin knockdown. (**a**) Validation of Tetherin knockdown using flow cytometry. Caco-2 control and knockdown cells (KO1, KO2) were stained using cell surface staining with a PE-conjugated anti-Tetherin antibody. (**b**) Relative Tetherin expression normalized to control cells. (**c**) Caco-2 control and KO cells were treated with IFNα for 24 h. Cells were fixed with 2% PFA, permeabilized and stained for total Tetherin expression. (**d**,**e**) Caco-2 control and Tetherin knockdown cells were infected with (**d**) SARS-CoV-2 ΔORF6-YFP or (**e**) SARS-CoV-2 ΔORF7-mNG at MOIs 0.05 and 0.04, respectively. Viral replication rates and spread, evident by fluorescent signal, were monitored for 72 h using live-cell imaging. Area of green (i.e., infected) cells over total cell area was quantified. Representative images are shown below the graph and indicate virus spread in control versus knockdown cells at the indicated time points (scale bar = 400 µm). (**f**,**g**) Quantification of the areas under the curve for (**f**) SARS-CoV-2 ΔORF6-YFP or (**g**) SARS-CoV-2 ΔORF7-mNG for different MOIs. (**d**–**g**) Mean values of three independent biological replicates conducted in technical triplicates are shown. Statistical significance was tested by one-way ANOVA with Sidak’s multiple comparisons (* *p* ≤ 0.05).

## Data Availability

All data generated and analyzed during this study are included in this published manuscript.

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
