# Peer review of "Tetherin Restricts SARS-CoV-2 despite the Presence of Multiple Viral Antagonists"

_viruses, 2023, doi:10.3390/v15122364_

Round 1
Reviewer 1 Report
Comments and Suggestions for Authors
Hagelauer et al. investigate the extent to which SARS-CoV and SARS-CoV-2 Spike and ORF7a proteins promote suppression of host Tetherin levels and function in human cell lines. The experiments are largely appropriately designed, and the manuscript is well written. The findings from this work are important, and complement other recent studies in the field.
I also appreciate that the authors provided accompanying uncropped blots with the manuscript.
I have a few comments as follows:
1.lines 99-100 - I am not sure surface staining is enough to validate the Tetherin KO cells. Western blotting for total Tetherin will be more appropriate here, especially since the authors already have the reagents to do this. In the absence of western blotting, combining intracellular and surface staining will be better than surface staining alone. Additionally, given that Tetherin is an ISG, having a condition where the cells have been treated with interferons would have been useful, to show that the knockout has worked.
2.lines 254-258 - The wording used here sounds too decisive, and I am not sure the blot is as convincing as the authors suggest. Without the proper controls, the appearance of an extra band in the ORF7a-expressing cells can just as easily be caused by degradation or cleavage. For example, the extra bands seen in the GFP-only cells on the Tetherin blot in figure 2e do not also indicate glycosylation.
3.line 277 - Given that figures 2a and 2b are assessing surface expression of Tetherin, and are discussed in section 3.1 of the text, perhaps the authors should consider moving them to the end of figure 1 and making them figure 1f and 1g, respectively.
4.line 277 - There is no mention of a APC-conjugated anti-Tetherin in the methods. Is this a typo? If it is an APC-conjugated anti-Tetherin, why was the PE-Tetherin antibody not used here?
5.lines 280-281 - The way it is written now, it is not immediately clear that the Tetherin plasmid was co-transfected with plasmids encoding the viral proteins. Perhaps the authors can revise it to something like 'HEK293T cells were co-transfected with a plasmid encoding human Tetherin, and indicated bicistronic plasmids encoding for SARS-CoV and SARS-CoV-2 S or ORF7a protein together with GFP from a bicistronic plasmid', for example.
6.line 297 - While the authors show colocalization of Tetherin with S and ORF7a, throughout the manuscript there is no mention of the particular fluorescent microscope used to acquire the images, including in the Methods section. I am sceptical of colocalization in this context without any data showing the three-dimentional volume of the cells such as by confocal microscopy.
7.line 328 - I suggest the authors use a line or something similar to more clearly indicate that the asterisks in figure 4c are for comparisons between Gal4 and Tetherin in the S bait-expressing cells. I think the authors should also indicate whether the comparison for the ORF7a bait-expressing cells is statistically significant or not, especially given the log10 scale used on the y axis.
8.line 355-356 - While I understand the authors' rationale for using bulk-cultures, I think using multiple monoclonal cultures of KO cells would have been a better approach as it will give a cleaner phenotype (due to total absence of Tetherin) while also ruling out potential off targets (by using multiple clones).
9.line 365 - In figure 5 (and section 3.4):
a.It would have been useful for the authors to additionally validate their CRISPR/Cas9 KO by western blotting, although I do understand that the KO may not be obvious in this case since pooled/bulk cells were used.
b.Using histograms in figure 5a in addition to the dotplots to show the KOs will make sense
c.What is the gating strategy used in 5a? It is not obvious from the figure legend or methods
d.Has the Tetherin staining actually worked in figure 5a? I would have expected a greater shift to the right in the control KO cells compared to the unstained control. As is, it looks like only 13.34% of control KO cells express Tetherin. I wonder if Tetherin expression in the different conditions could be better demonstrated in cells treated with interferons, given that Tetherin is an ISG.
e.Given that the authors already have the flow cytometry setup, it would have made sense for them to also use it to assess virus replication in the knockouts. That way, they can look for replication in the Tetherin KO cells and complement the data from the live cell imaging.
f.It would have been useful here for the authors to assess whether there are any changes in the surface and total Tetherin in the context of infection, to complement their data from figures 1 and 2. Given that the data from figure 1 shows no effect in Caco-2 cells, it is perhaps not surprising that viral replication was increased in the KO cells. Perhaps A549 or HEK293T cells expressing the viral entry receptor would have been more appropriate, in hindsight.
10.lines 401-402 - Without the proper controls, I am not sure the data in figure 2c shows this.
11.line 413 - I think the word 'withhold' as used here is not appropriate. As is, the sentence implies that the virus is 'not using' additional mechanisms to blunt the interferon response, which is not the case. Perhaps the word 'use' can be used in its stead.
12.I wonder if the authors can speculate in the discussion as to the possible explanation for the cell type-dependent differences in the phenotypes observed in figure 1.
Comments on the Quality of English Language
Manuscript was well-written
Author Response
Hagelauer et al. investigate the extent to which SARS-CoV and SARS-CoV-2 Spike and ORF7a proteins promote suppression of host Tetherin levels and function in human cell lines. The experiments are largely appropriately designed, and the manuscript is well written. The findings from this work are important, and complement other recent studies in the field.
I also appreciate that the authors provided accompanying uncropped blots with the manuscript.
We very much appreciate the positive feedback from Reviewer 1. As detailed below, we addressed all the comments.
I have a few comments as follows:
1.lines 99-100 - I am not sure surface staining is enough to validate the Tetherin KO cells. Western blotting for total Tetherin will be more appropriate here, especially since the authors already have the reagents to do this. In the absence of western blotting, combining intracellular and surface staining will be better than surface staining alone. Additionally, given that Tetherin is an ISG, having a condition where the cells have been treated with interferons would have been useful, to show that the knockout has worked.
Following this recommendation, we assessed total tetherin levels in our KO cells with IFN treatment. The data is now in the new panel C Fig.5.
2.lines 254-258 - The wording used here sounds too decisive, and I am not sure the blot is as convincing as the authors suggest. Without the proper controls, the appearance of an extra band in the ORF7a-expressing cells can just as easily be caused by degradation or cleavage. For example, the extra bands seen in the GFP-only cells on the Tetherin blot in figure 2e do not also indicate glycosylation.
We agree and changed the wording to a more careful interpretation of the data.
3.line 277 - Given that figures 2a and 2b are assessing surface expression of Tetherin, and are discussed in section 3.1 of the text, perhaps the authors should consider moving them to the end of figure 1 and making them figure 1f and 1g, respectively.
When assembling the data into figures we indeed considered the structure as suggested by the reviewer. However, after discussion with all authors and other colleagues we decided to keep the actual logical flow, as Fig-1 shows cell surface regulation of endogenous tetherin whereas figure 2 exclusively focuses on S and ORF7A-mediated effects on 293T cells transfected to express tetherin.
4.line 277 - There is no mention of a APC-conjugated anti-Tetherin in the methods. Is this a typo? If it is an APC-conjugated anti-Tetherin, why was the PE-Tetherin antibody not used here?
Depending on the experimental setting we used APC or PE conjugated clones on anti-Tetherin. In the revised version of the manuscript we detail this in the M&M section.
5.lines 280-281 - The way it is written now, it is not immediately clear that the Tetherin plasmid was co-transfected with plasmids encoding the viral proteins. Perhaps the authors can revise it to something like 'HEK293T cells were co-transfected with a plasmid encoding human Tetherin, and indicated bicistronic plasmids encoding for SARS-CoV and SARS-CoV-2 S or ORF7a protein together with GFP from a bicistronic plasmid', for example.
We agree and rewrote the paragraph accordingly.
6.line 297 - While the authors show colocalization of Tetherin with S and ORF7a, throughout the manuscript there is no mention of the particular fluorescent microscope used to acquire the images, including in the Methods section. I am sceptical of colocalization in this context without any data showing the three-dimentional volume of the cells such as by confocal microscopy.
We did sectioning of cells however given the obvious colocalization of tetherin with ORF7a and S we decided to keep data presentation simple with 2D confocal images and line plots to show colocalization of the proteins. Details of the equipment used was added to the M&M section.
7.line 328 - I suggest the authors use a line or something similar to more clearly indicate that the asterisks in figure 4c are for comparisons between Gal4 and Tetherin in the S bait-expressing cells. I think the authors should also indicate whether the comparison for the ORF7a bait-expressing cells is statistically significant or not, especially given the log10 scale used on the y axis.
We agree and added the line as well as the comparison to ORF7a bait expressing cells.
8.line 355-356 - While I understand the authors' rationale for using bulk-cultures, I think using multiple monoclonal cultures of KO cells would have been a better approach as it will give a cleaner phenotype (due to total absence of Tetherin) while also ruling out potential off targets (by using multiple clones).
Even though there is a good rationale to do KO experiments with monoclonal cell lines, we decided against this strategy when designing the experiments. The rational was, that in our hands there is often differential morphology and growth kinetics of monoclonal cells, likely due to unpredictable off target effects. Retrospective, we could have followed up on testing bulk as well as monoclonal KO cells and we will likely follow up on this routine in further experiments. However for this work, we feel that redoing all the experiments in yet-to-establish monoclonal KO cells is beyond the current scope.
9.line 365 - In figure 5 (and section 3.4):
a.It would have been useful for the authors to additionally validate their CRISPR/Cas9 KO by western blotting, although I do understand that the KO may not be obvious in this case since pooled/bulk cells were used.
We agree and performed total tetherin level assessment by intracellular flow cytometry staining (new Fig. 5 C).
b.Using histograms in figure 5a in addition to the dotplots to show the KOs will make sense
In the revised version we also use histograms to show the Tetherin KOs.
c.What is the gating strategy used in 5a? It is not obvious from the figure legend or methods
We detailed the gating strategy in the methods in the revised manuscript version.
d.Has the Tetherin staining actually worked in figure 5a? I would have expected a greater shift to the right in the control KO cells compared to the unstained control. As is, it looks like only 13.34% of control KO cells express Tetherin. I wonder if Tetherin expression in the different conditions could be better demonstrated in cells treated with interferons, given that Tetherin is an ISG.
We thank the reviewer and followed up on the suggestion. Indeed, the KO is still clearly visible when cells were treated with IFN and is up to 50% down (new Fig. 5 C).
e.Given that the authors already have the flow cytometry setup, it would have made sense for them to also use it to assess virus replication in the knockouts. That way, they can look for replication in the Tetherin KO cells and complement the data from the live cell imaging.
We agree and we have paralleled that for some of the biological replicates. However, as we witnessed similar results when monitoring spread of the reporter virus by fluorescence microscopy as compared to titrating virus in cell culture supernatants and both information is redundant we decided to keep the live cell imaging setup.
f.It would have been useful here for the authors to assess whether there are any changes in the surface and total Tetherin in the context of infection, to complement their data from figures 1 and 2. Given that the data from figure 1 shows no effect in Caco-2 cells, it is perhaps not surprising that viral replication was increased in the KO cells. Perhaps A549 or HEK293T cells expressing the viral entry receptor would have been more appropriate, in hindsight.
As suggested we performed infections in ACE2-expressing A549 cells but had very low infection rates for thus far unknown reasons while mirroring the results in Caco2 cells. We did not test 293T. Overall, we then decided to perform experiments in a cell line that is naturally permissive for SARS-CoV-2 infection without the necessity to overexpress its entry receptors (as it is necessary for A549 or 293T). Overall, we agree that replicating these experiments in primary cells would be the most relevant, which is difficult to realize due to the high divergence of permissivity towards infection when using SAEC or NHBE in conjunction with Crispr/Cas9 KO. Nevertheless, results in Caco2 support the main conclusion of our study that Tetherin is capable of restricting overall replication and spread of SARS-CoV-2 even if antagonistic function of variable viral proteins have been described.
10.lines 401-402 - Without the proper controls, I am not sure the data in figure 2c shows this.
We agree and modified this statement.
11.line 413 - I think the word 'withhold' as used here is not appropriate. As is, the sentence implies that the virus is 'not using' additional mechanisms to blunt the interferon response, which is not the case. Perhaps the word 'use' can be used in its stead.
We modified the sentence as suggested by the Reviewer.
12.I wonder if the authors can speculate in the discussion as to the possible explanation for the cell type-dependent differences in the phenotypes observed in figure 1.
Expression levels might be an issue, as 293Ts have very low tetherin expression. However as this is highly speculative we decided not to put an emphasis on this particular aspect.
Reviewer 2 Report
Comments and Suggestions for Authors
The work is extremely interesting and innovative. The results are promising and encourage further research. The authors demonstrated extensive knowledge and in-depth understanding of the presented topic. As a reviewer, I am obliged to point out shortcomings that, in my opinion, are minor and should be easy to remove. Congratulations on conducting and writing such an excellent piece of work. Below are my comments:
- the abstract lacks a brief description of the methodology.
- the introduction is written clearly and is understandable, but in my opinion, the initial section about the virus's structure, which is commonly known to interested scientists, is unnecessary.
- there is also a lack of a brief description of the applied statistical methods, from analysis of variance and distribution assessments to specific comparisons. It is hard to believe that all It is hard to believe that all obtained data were normally distributed
- the discussion is logically organized but is too condensed, making it difficult to read with many mental shortcuts. Certainly, just dividing the text into paragraphs would make it easier to read.
Author Response
The work is extremely interesting and innovative. The results are promising and encourage further research. The authors demonstrated extensive knowledge and in-depth understanding of the presented topic. As a reviewer, I am obliged to point out shortcomings that, in my opinion, are minor and should be easy to remove. Congratulations on conducting and writing such an excellent piece of work. Below are my comments:
We thank the reviewer for this very positive feedback on our study.
- the abstract lacks a brief description of the methodology.
We added a few sentences to the abstract about the methodology used.
- the introduction is written clearly and is understandable, but in my opinion, the initial section about the virus's structure, which is commonly known to interested scientists, is unnecessary.
As recommended, and also mentioned by Reviewer 3, we streamlined the introduction and shortened the initial paragraph to keep focused on the aim of the study.
- there is also a lack of a brief description of the applied statistical methods, from analysis of variance and distribution assessments to specific comparisons. It is hard to believe that all It is hard to believe that all obtained data were normally distributed
We mention the statistical test used in the respective figure legends. In the revised version, we added a paragraph to the Material&Methods section to briefly describe the applied statistical methods.
- the discussion is logically organized but is too condensed, making it difficult to read with many mental shortcuts. Certainly, just dividing the text into paragraphs would make it easier to read.
We broke up the discussion into paragraphs to ease reading.
Reviewer 3 Report
Comments and Suggestions for Authors
This paper is well written and somewhat increases nuance on a subtle and elegant way by which SARS-CoV-2 ( and other viruses) evades the IFN- tetherin control of virus exit from infected cells. Methodology is adequate to the study questions. I have only two points for the Authors: 1. The Introduction could be shortened, particularly its initial paragraph: most of this part is wery well known and dilutes from the focal aim of the study; 2. the results are of obvious interest but the differences are often to the limit or below the statistical confidence. I suggest the Authors write down a note of study limitations regarding this in the Discussion section .
Comments on the Quality of English Language
Good English, indeed
Author Response
This paper is well written and somewhat increases nuance on a subtle and elegant way by which SARS-CoV-2 ( and other viruses) evades the IFN- tetherin control of virus exit from infected cells. Methodology is adequate to the study questions. I have only two points for the Authors: 1. The Introduction could be shortened, particularly its initial paragraph: most of this part is wery well known and dilutes from the focal aim of the study; 2. the results are of obvious interest but the differences are often to the limit or below the statistical confidence. I suggest the Authors write down a note of study limitations regarding this in the Discussion section
We thank the reviewer for the positive assessment of our work.
As recommended, and also mentioned by Reviewer 2, we streamlined the introduction and shortened the initial paragraph to keep focused on the aim of the study.
Furthermore, we agree that especially S and ORF7A mediated cell surface regulation is subtle, even though statistically significant and it is questionable if this has any biological relevance. We mention and discuss this aspect in the results and discussion section.